# Quantum engineering of a synthetic thermal bath for bosonic atoms in a one-dimensional optical lattice via Markovian feedback control

Ling-Na Wu* and André Eckardt†

Institut für Theoretische Physik, Technische Universität Berlin,
Hardenbergstraße 36, Berlin 10623, Germany

* lingna.wu@tu-berlin.de ,    † eckardt@tu-berlin.de

## Abstract

We propose and investigate a scheme for engineering a synthetic thermal bath for a bosonic quantum gas in a one-dimensional optical lattice based on Markovian feedback control. The performance of our scheme is quantified by the fidelity between the steady state of the system and the effective thermal state. For double-well and triple-well systems with non-interacting particles, the steady state is found to be an exact thermal state, which is attributed to the fact that the transfer rates between all pairs of coupled eigenstates satisfy detailed balance condition. The scenario changes when there are more lattice sites, where the detailed balance condition does not hold any more, but remains an accurate approximation. Remarkably, our scheme performs very well at low and high temperature regimes, with the fidelity close to one. The performance at intermediate temperatures (where a crossover into a Bose condensed regime occurs) is slightly worse, and the fidelity shows a gentle decrease with increasing system size. We also discuss the interacting cases. In contrast to the non-interacting cases, the scheme is found to perform better at a higher temperature. Another difference is that the minimal temperature that can be engineered is nonzero and increases with the interaction strength.

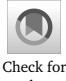

# 1   Introduction

Atomic quantum gases in optical lattice constitute a unique experimental platform for studying quantum many-body systems due to their high controllability and good isolation from the environments [1]. The latter offers a unique opportunity to experimentally study coherent nonequilibrium dynamics of many-body systems [2]. It however also blocks the study of interesting non-equilibrium properties of open quantum systems. This topic has attracted tremendous interest in recent years, including non-equilibrium steady state in driven-dissipative systems [3–14], dissipative phase transition where the steady state exhibits various nonequilibrium phases and phase transitions due to the competition of unitary and dissipative dynamics [15–24], the effect of dissipation on many-body localization [25–40], and dissipation engineering where the coulping between the system and the reservoir is designed to steer the system dynamics for a prespecified target [41–49], just to name a few.

In order to study non-equilibrium steady states and dynamics of open systems in atomic quantum gases, one has to artificially synthesize an environment. Dephasing noise has been implemented in experiments with ultracold atoms via the off-resonant scattering of lattice photons [31, 50]. Another approach to engineer a bath is introducing a second species of atoms [51, 52] which plays the role of environment. There are also experiments which use a second hyperfine state to serve as a bath [53]. Here, we consider the quantum engineering of a synthetic thermal bath via Markovian feedback control [54, 55]. As an important measurement-based feedback method, Markovian feedback control has been applied to various problems, including the stabilization of arbitrary one-qubit quantum states [56, 57], the manipulation of quantum entanglement between two qubits [58–61], as well as optical and spin squeezing [62–64].

In our previous work [65], we have shown that Markovian feedback control can be used to prepare single targeted eigenstates of a bosonic quantum gas in an optical lattice. Here, we will use a similar feedback scheme to realize the engineering of a thermal bath. Our paper is organized as follows: a brief introduction to the Markovian feedback theory is given in Section 2, which is followed by the description of our model in Section 3. The main results are shown in Section 4. Here we present the basic idea of our scheme and focus on the non-interacting case. For illustration, we start with the discussion of small systems with two sites in Section 4.1, three sites in Section 4.2, and four sites in Section 4.3. We then discuss the general case in Section 4.4. This is followed by a discussion of system-size dependence in Section 4.5, where kinetic theory is exploited to treat large systems. The interacting case is discussed in Section 5 before a summary of the main results in Section 6 to conclude.

## 2  Markovian feedback control

Our scheme is based on the Markovian feedback master equation (ME) [54,55]. Here we give a brief introduction to it. Suppose a system described by Hamiltonian $H$ is under a homodyne measurement described by operator $c$, its dynamics is then governed by the stochastic master equation (SME) [54,55] ($\hbar = 1$ hereafter),

$$d\rho_c = -i[H,\rho_c]dt + \mathcal{D}[c]\rho_c dt + \mathcal{H}[c]\rho_c dW\,, \tag{1}$$

with

$$\begin{aligned}
\mathcal{H}[c]\rho &:= c\rho + \rho c^\dagger - \mathrm{Tr}[(c+c^\dagger)\rho]\rho\,,\\
\mathcal{D}[c]\rho &:= c\rho c^\dagger - \frac{1}{2}(c^\dagger c\rho + \rho c^\dagger c)\,.
\end{aligned} \tag{2}$$

Here $\rho_c$ denotes the quantum state conditioned on the measurement result,

$$I_{\mathrm{hom}} = \mathrm{Tr}[(c+c^\dagger)\rho] + \xi(t)\,, \tag{3}$$

with $\xi(t) = dW/dt$ and $dW$ being the standard Wiener increment with mean zero and variance $dt$. By using the information acquired from the measurements, one can introduce feedback control to the system which allows to steer the system's dynamics to achieve desired effects.

There are various strategies to implement the feedback control [66]. Here we consider a direct feedback scheme, where a signal-dependent, i.e. conditional, feedback term $I_{\mathrm{hom}}F$ is added to the Hamiltonian. Such a feedback is assumed to be instantaneous, namely, the delay time between the measurement and the application of the control field is small compared to the typical timescales of the system. For cold atoms in cavity (which is a potential experimental setup to implement our scheme), the typical time scales (such as tunneling time) are on the order of milliseconds [69]. Hence, a control on the higher kHz scale is sufficient, which can be achieved easily using digital signal processors. This assumption ensures the Markovianity of the dynamical description. According to Markovian feedback control theory [54,55], the system is then governed by the feedback-modified SME

$$d\rho_c = -i[H+H_{\mathrm{fb}},\rho_c]dt + \mathcal{D}[A]\rho_c dt + \mathcal{H}[A]\rho_c dW\,, \tag{4}$$

with collapse operator

$$A = c - iF\,, \tag{5}$$

and feedback-induced term $H_{\mathrm{fb}} = \frac{1}{2}(c^\dagger F + Fc)$. By comparing Eqs. (4) and (1), one can see that the effect induced by the feedback loop is replacing the collapse operator $c$ by $A$ and adding an extra term $H_{\mathrm{fb}}$ to the Hamiltonian.

By taking the ensemble average of the possible measurement outcomes, we arrive at the feedback-modified ME [54,55]

$$\frac{d\rho}{dt} = -i[H+H_{\mathrm{fb}},\rho] + \mathcal{D}[A]\rho \equiv \mathcal{L}\rho\,. \tag{6}$$

Note that we have assumed perfect detection with efficiency $\eta = 1$. The following discussions will be focused on the steady state of this ME unless stated otherwise.

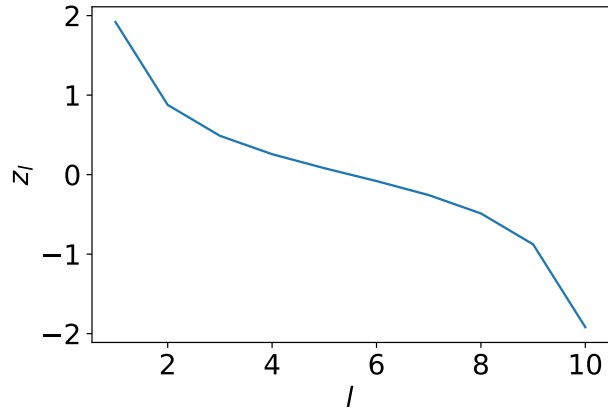

Figure 1: The coefficient $z_l$ in the measurement operator, see Eq. (9), for $M = 10$.

## 3 Model

The system under consideration is a bosonic quantum gas of $N$ atoms in a one-dimensional chain with open boundary condition. It is described by the Bose-Hubbard Hamiltonian

$$H = -J \sum_{l=1}^{M-1} (a_l^\dagger a_{l+1} + a_{l+1}^\dagger a_l) + \frac{U}{2} \sum_{l=1}^{M} n_l(n_l - 1), \tag{7}$$

where $a_l$ ($a_l^\dagger$) annihilates (creates) a particle on site $l$ and $n_l = a_l^\dagger a_l$ counts the particle number on site $l$, with $\sum_l n_l = N$. The first term in (7) describes tunneling between nearest neighbor sites with rate $J$, and the second term denotes on-site interaction with strength $U$.

We are interested in the quantum simulation of the coupling of this system to a thermal bath based on Markovian feedback control. For this purpose, we consider the measurement operator [65]

$$c = \sqrt{\gamma} \sum_{l=1}^{M} z_l n_l, \tag{8}$$

which is a weighted sum of the on-site occupations with

$$z_l = \frac{g_{l+1} - g_{l-1}}{g_l}, \quad g_l = \sqrt{\frac{2}{M+1}} \sin\left(\frac{\pi l}{M+1}\right), \tag{9}$$

and feedback operator

$$F = -i\sqrt{\gamma}\lambda \sum_{l=1}^{M-1} (a_l^\dagger a_{l+1} - a_{l+1}^\dagger a_l), \tag{10}$$

which describes nearest neighbor tunneling with a complex tunneling rate. Here, $\gamma$ denotes measurement strength and $\lambda$ is a free parameter to be optimized. Figure 1 shows the coefficient $z_l$ in the measurement operator for a system with $M = 10$ sites. The measurement (8) can be implemented via homodyne detection of the off-resonant scattering of structured probe light from the atoms [67–69]. The detection efficiency can be enhanced by placing the system inside an optical cavity [69–72], so that via the Purcell effect [73] the photons are scattered predominantly into one cavity mode. The feedback (10) can be realized by accelerating the lattice [65,74].

We will focus on the weak measurement case (with the measurement strength $\gamma$ much smaller than the typical energy scale of the system) so that the effect of the feedback-induced

term $H_{\text{fb}}$ in Eq. (6) is small compared to the system Hamiltonian $H$ and the steady state is diagonal in the eigenbasis of the system. To quantify the performance of our scheme, we use the fidelity between the steady state $\rho_{\text{ss}}$ and the effective thermal state $\rho_T$,

$$f = \text{tr}\sqrt{\sqrt{\rho_T}\rho_{\text{ss}}\sqrt{\rho_T}}, \tag{11}$$

where

$$\rho_T = e^{-\beta H}/\text{tr}(e^{-\beta H}), \tag{12}$$

with inverse temperature $\beta = 1/(k_B T)$ (the Bolzmann constant $k_B$ is set to be 1 hereafter).

## 4 Non-interacting case

Let us start with the non-interacting case, i.e., $U = 0$. The $j$th single-particle eigenstate is given by

$$|j\rangle = \sum_{l=1}^{M} g_l^{(j)}|l\rangle, \quad g_l^{(j)} = \sqrt{\frac{2}{M+1}}\sin\left(\frac{\pi j l}{M+1}\right), \tag{13}$$

with eigenenergy $E_j = -2J\cos\frac{j\pi}{M+1}$. Note that '$g_l$' in the coefficient $z_l$ of the measurement operator (8) is exactly $g_l^{(1)}$.

We have shown in Ref. [65] that for $\lambda = 1$, the ground state of the non-interacting system $|1\rangle^{\otimes N}$ (with all particles occupying the single-particle ground state $|1\rangle$) is the unique dark state of the collapse operator $A = c - iF$, i.e., $A|1\rangle^{\otimes N} = 0$. Thus, the steady state of the system is the ground state, which corresponds to a zero-temperature state. If we set $\lambda = -1$, then the steady state becomes the highest excited state, which corresponds to a negative-temperature state. For $\lambda = 0$, i.e., when there is no feedback control, the collapse operator is given by the measurement operator $c$, which is hermitian, the steady state then becomes the maximally mixed state, i.e., an equal mixture of all the eigenstates, which corresponds to an infinite-temperature state.

From these results, one might wonder how the steady state of the so-controlled system looks, when varying $\lambda$ continuously from 1 to $-1$. Quite remarkably, we find that not only the energy of the steady state continuously increases from the ground-state energy to the energy of the most excited state, but also that the steady state is to very good approximation given by a thermal state, whose inverse temperature smoothly changes from $\infty$ to $-\infty$. Thus, solely by varying $\lambda$ thermal steady states of arbitrary temperature can be prepared.

To illustrate this point, we first study small systems with $M = 2$ (Section 4.1), 3 (Section 4.2) and 4 (Section 4.3), and then discuss the case of general $M$ in Section 4.4. Note that for the non-interacting cases, the mapping between $\lambda$ and the temperature $T$ of the effective thermal bath does not depend on the particle number $N$. Hence, we will focus on the single particle case for the discussion of $\lambda$-$T$ mapping in Sections 4.1-4.4. The system-size dependence of fidelity is discussed in Section 4.5, where kinetic theory is introduced in order to treat large systems.

Before jumping to the discussion of our scheme, let us first recapitulate the steady state of a system which is weakly coupled to a bath [75]. The dynamics of the occupation probabilities in the eigenstates, $p_i \equiv \langle i|\rho|i\rangle$, decouples from the off-diagonal elements of the density operator which decay as one approaches the steady state, and is described by the Pauli rate equation,

$$\dot{p}_i = \sum_j \left(p_j R_{ij} - p_i R_{ji}\right), \tag{14}$$

where $R_{ij}$ denotes the transfer rate from eigenstate $|j\rangle$ to $|i\rangle$. The terms of the sum correspond to the net probability flux from states $|j\rangle$ to state $|i\rangle$. The steady state is obtained by setting $\dot{p}_i = 0$, i.e.,

$$\sum_j \left( p_j R_{ij} - p_i R_{ji} \right) = 0 \,. \tag{15}$$

If the bath is a thermal bath of temperature $T$, the transfer rates satisfy detailed balance condition,

$$\frac{R_{ij}}{R_{ji}} = e^{-(E_i - E_j)/T} \,. \tag{16}$$

This condition implies that the steady state, obtained by solving Eq. (15), is given by the Gibbs state with $p_i = \mathcal{Z}^{-1} e^{-E_i/T}$ and $\mathcal{Z} = \sum_i e^{-E_i/T}$. For this equilibrium state, the sum on the right-hand side of Eq. (15) vanishes term by term. Thus, the net probability flux between two states $|i\rangle$ and $|j\rangle$ vanishes. This is the property of detailed balance, which is characteristic for the thermodynamic equilibrium.

## 4.1 Two sites

For a double-well system with $M = 2$, we can write out the collapse operator $A = c - iF$ in the eigenbasis $\{|g\rangle, |e\rangle\}$ as

$$A = \sqrt{\gamma} \begin{pmatrix} 0 & 1+\lambda \\ 1-\lambda & 0 \end{pmatrix} = \sqrt{\gamma}(1+\lambda)|g\rangle\langle e| + \sqrt{\gamma}(1-\lambda)|e\rangle\langle g| \,. \tag{17}$$

By setting $\lambda = 1$, the second term disappears and the ground state $|g\rangle$ becomes the dark state of $A$, i.e., $A|g\rangle = 0$. In turn, the steady state will be the ground state $|g\rangle$. Likewise, by setting $\lambda = -1$, the first term disappears and the steady state will be the excited state $|e\rangle$. For $-1 < \lambda < 1$, both terms exist and the steady state will be a mixed state.

From Eq. (17), we can see that the transfer rate from the ground state $|g\rangle$ to the excited state $|e\rangle$ is $R_{eg} = |\langle e|A|g\rangle|^2 = \gamma(1-\lambda)^2$, and the transfer rate from the excited state $|e\rangle$ to the ground state $|g\rangle$ is $R_{ge} = |\langle g|A|e\rangle|^2 = \gamma(1+\lambda)^2$. Suppose the system is coupled to a thermal bath at temperature $T$, the ratio between these two rates should satisfy the detailed balance condition

$$\frac{R_{eg}}{R_{ge}} = e^{-(E_e - E_g)/T} \,, \tag{18}$$

which implies a thermal probability distribution. Hence, our model can be used to mimic the coupling of the double-well system to a thermal bath at temperature $T$ by imposing

$$\frac{(1-\lambda)^2}{(1+\lambda)^2} = e^{-2J/T} \,. \tag{19}$$

This leads to a mapping between the free parameter $\lambda$ in our feedback scheme (10) and the temperature of the effective thermal bath,

$$T_2 = \frac{1}{\log\left(\frac{1+\lambda}{1-\lambda}\right)} J \,. \tag{20}$$

Note that although for a two-level system every steady state which is diagonal in the eigenbasis trivially corresponds to a thermal state with some temperature, our results become non-trivial for larger system sizes, as discussed below.

## 4.2 Three sites

For a triple-well system with $M = 3$, the collapse operator $A = c - iF$ in the eigenbasis reads

$$A = \sqrt{\gamma} \begin{pmatrix} 0 & 1+\lambda & 0 \\ 1-\lambda & 0 & 1+\lambda \\ 0 & 1-\lambda & 0 \end{pmatrix}. \tag{21}$$

If the system is coupled to a thermal bath, each pair of the transfer rates between two coupled eigenstates should satisfy the detailed balance condition. In the special case considered here (21), we have two identical pairs of transfer rates, and the detailed balance condition can be satisfied by setting

$$\frac{R_{ij}}{R_{ji}} = e^{-(E_i - E_j)/T}, \tag{22}$$

with the transfer rate $R_{ij} = |\langle i|A|j\rangle|^2$ and energy $E_j = -2J\cos\frac{j\pi}{M+1} = \{-\sqrt{2}J, 0, \sqrt{2}J\}$. The temperature of the effective thermal bath is related to the free parameter $\lambda$ as

$$T_3 = \frac{J}{\sqrt{2}\log\left(\frac{1+\lambda}{1-\lambda}\right)} = \frac{T_2}{\sqrt{2}}. \tag{23}$$

## 4.3 Four sites

For a system with four sites, the collapse operator $A = c - iF$ in the eigenbasis reads

$$A = \sqrt{\gamma} \begin{pmatrix} 0 & A_{12,+} & 0 & A_{14,+} \\ A_{12,-} & 0 & A_{23,+} & 0 \\ 0 & A_{23,-} & 0 & A_{34,+} \\ A_{14,-} & 0 & A_{34,-} & 0 \end{pmatrix}, \tag{24}$$

where

$$\begin{aligned} A_{12,\pm} &= \frac{2}{\sqrt{5}}(1\pm\lambda), \\ A_{14,\pm} &= \frac{3-\sqrt{5}}{2\sqrt{5}}(1\pm\lambda), \\ A_{23,\pm} &= \frac{4-2\sqrt{5}}{2\sqrt{5}} + \frac{3+\sqrt{5}}{2\sqrt{5}}(1\pm\lambda), \\ A_{34,\pm} &= \frac{2}{\sqrt{5}}(1\pm\lambda) = A_{12,\pm}. \end{aligned} \tag{25}$$

In this case, we have four pairs of transfer rates and it is impossible for all of them to satisfy the detailed balance condition. Neverthethess the steady state is found to be close to a thermal state. As shown in Fig. 2 (a) and (b), the fidelity between the steady state and the corresponding thermal state is very close to one over the whole parameter regime. Here, the inverse temperature of the thermal state is fixed by optimizing the fidelity and is shown in (c). We also compare the distribution of the steady state (solid lines) and the thermal state (dashed lines) in the eigenbasis in (d). The overall behavior is found to be quite similar. Note that the results are symmetric with respect to $\lambda = 0$. In the following discussion, we will focus on the case with $\lambda > 0$.

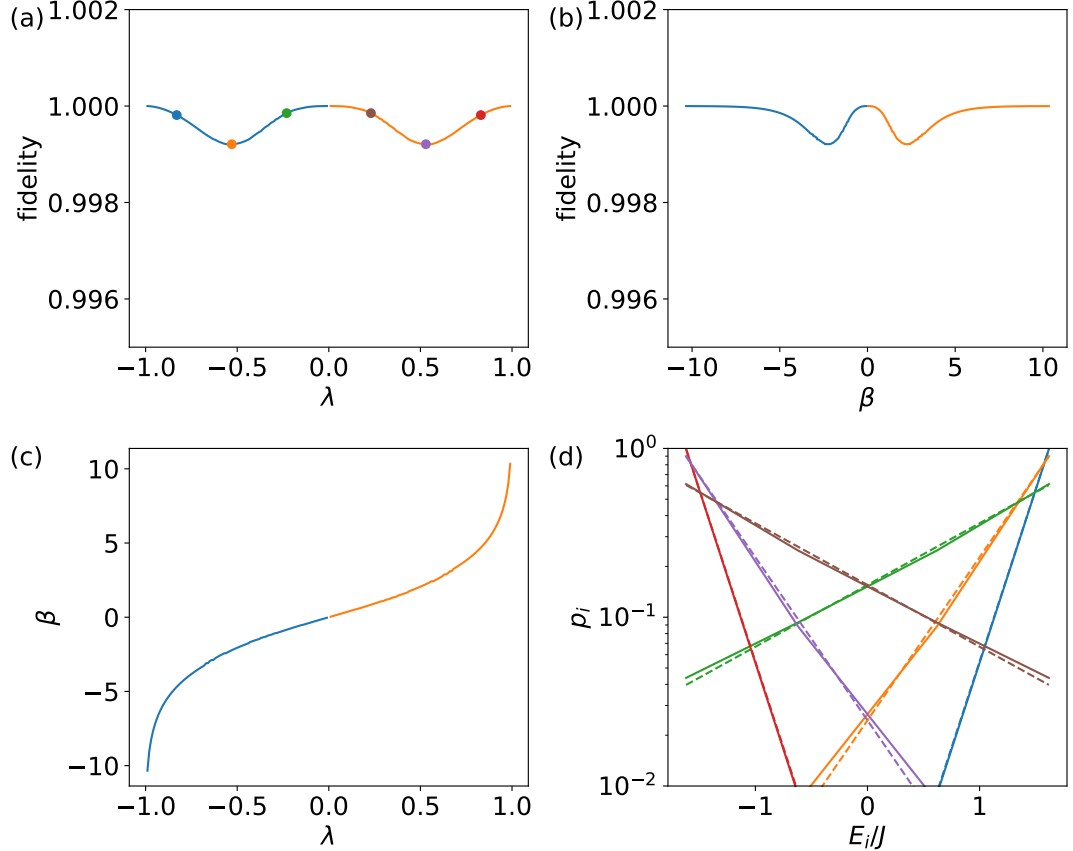

Figure 2: The fidelity between the steady state and the corresponding thermal state as a function of (a) the free parameter $\lambda$ and (b) the inverse temperature $\beta$. The inverse temperature of the thermal state is fixed by optimizing the fidelity and is shown in (c). (d) compares the distribution of the steady state (solid lines) and the corresponding thermal state (dashed lines) on the eigenbasis of the system at various $\lambda$ [marked by circles in (a)]. The parameters are $M = 4$, $N = 1$, $U = 0$, and $\gamma = 0.001J$.

## 4.4 General $M$

For general $M$, we cannot find an exact expression between $\lambda$ and the temperature of the effective thermal bath $T$, as in the double-well and triple-well case [Eqs. (20) and (23)]. Numerically, one can fix $T$ by optimizing the fidelity between the steady state and a thermal state with the temperature as a control parameter. A typical $\lambda$-$T$ mapping is shown in Fig. 3 (solid line). As expected, a smaller $\lambda$ corresponds to a higher temperature.

At some parameter regime, we can get an approximated expression for the $\lambda$-$T$ mapping. For $\lambda \sim 1$ (low temperatures), the main population is concentrated in the first two eigenstates. The distribution satisfies

$$R_{12}p_2 - \sum_j R_{j1}p_1 = 0 \,. \tag{26}$$

By mapping this distribution to a thermal distribution,

$$\frac{p_1}{p_2} = e^{-(E_1 - E_2)/T} \,, \tag{27}$$

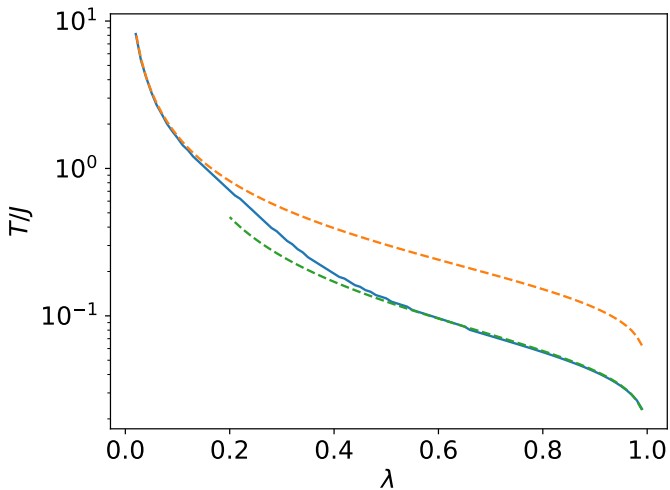

Figure 3: The temperature of the effective thermal bath as a function of $\lambda$. The blue line denotes the result from numerical optimization where $T$ is optimized to give the highest fidelity between the steady state and the corresponding thermal state. The green dashed line is the approximated result of Eq. (33) for low temperatures, and the orange line is the approximated result of Eq. (34) for high temperatures. The parameters for numerical simulations are $N = 1$, $M = 10$, $U = 0$, $\gamma = 0.001J$.

we get

$$T = \frac{E_2 - E_1}{\log(p_1/p_2)} = \frac{E_2 - E_1}{\log(R_{12}/\sum_j R_{j1})}. \tag{28}$$

By using $R_{ij} = |\langle i|A|j\rangle|^2$, with $A = c - iF$, $c = \sqrt{\gamma}\sum_l z_l|l\rangle\langle l|$, $F = -i\sqrt{\gamma}\lambda\sum_l(|l\rangle\langle l+1|-|l+1\rangle\langle l|)$, Eqs. (9) and (13), one can show that for odd $j$, $R_{1j} = R_{j1} = 0$, and for even $j$, $R_{1j} = \gamma A_{1j,+}^2$, $R_{j1} = \gamma A_{1j,-}^2$, where

$$A_{1j,\pm} = f(j,\alpha)\frac{1 \pm \lambda}{M+1}, \tag{29}$$

with

$$f(j,\alpha) = \frac{2\sin\alpha\sin(j\alpha)}{\sin[(j+1)\alpha/2]\sin[(j-1)\alpha/2]}, \tag{30}$$

and $\alpha = \pi/(M+1)$. And the energy gap reads

$$\Delta E = E_2 - E_1 = 2J\left[\cos\alpha - \cos(2\alpha)\right]. \tag{31}$$

For large $M$, we have $\sin\alpha \simeq \alpha$, and thus $f(j,\alpha) \simeq 8j/(j^2-1)$. This gives

$$R_{12} \simeq \left(\frac{16}{3}\right)^2 \gamma\frac{(1+\lambda)^2}{(M+1)^2},$$

$$\sum_j R_{j1} \simeq 38\gamma\frac{(1-\lambda)^2}{(M+1)^2}. \tag{32}$$

Substitution of it into Eq. (28) gives

$$T \simeq \frac{\Delta E}{\log\left[\frac{14(1+\lambda)^2}{19(1-\lambda)^2}\right]}. \tag{33}$$

For $\lambda \sim 0$ (high temperatures), the mapping between $\lambda$ and $T$ is found to be well approximated by

$$T \simeq \frac{T_2}{\sqrt{M-1}} = \frac{J}{\sqrt{M-1}\log\left(\frac{1+\lambda}{1-\lambda}\right)}. \tag{34}$$

In Fig. 3, we compare the numerical results (solid lines) with the above approximated expressions [Eqs. (33) and (34)] (dashed lines) and find good agreements in the corresponding parameter regimes.

## 4.5 System-size dependence

Now we investigate the system-size dependence of the fidelity. Figure 4(a) shows the fidelity for various particle number $N$ with lattice site number $M = 4$. We can see that overall the fidelity is very high, especially at low and high temperatures. The fidelity at the intermediate temperature regime is a bit lower, and shows a gentle decrease as $N$ increases [see Fig. 4(b)]. In Figs. 4(c) and (d), we fix the particle number at $N = 1$ and investigate the dependence of the fidelity on the lattice site number $M$. The fidelity is found to decrease with increasing $M$, as $1.12M^{-0.04}$ from curve fitting. It is not surprising to see this behavior, as when the system size increases, there are more transfer rates, and the approximation to the detailed balance condition will become worse. Nevertheless, fidelities larger than 90, 95, 99 percent are found for systems of size $M \lesssim 150$, $M \lesssim 40$, and $M \lesssim 10$, respectively.

All of the above discussions are based on the numerical calculation of steady state of the ME (6), which is obtained by the exact diagonalization of the Liouvillian superoperator $\mathcal{L}$, with $\mathcal{L}\rho = -i[H + H_{\text{fb}}, \rho] + \mathcal{D}[A]\rho$. For a system with $N$ particles and $M$ sites, the dimension of the Hilbert space is $D = (N + M - 1)!/N!/(M - 1)!$, and the Liouvillian superoperator $\mathcal{L}$ is a $D^2$ by $D^2$ matrix. For instance, for $M = 4$ and $N = 8$, $D = 330$, thus we need to diagonalize a 108900 by 108900 matrix. This simple example shows that it is hard to treat large systems by using the exact diagonalization approach.

To circumvent this problem, we resort to kinetic theory for the mean occupations in the single-particle eigenstate $\langle n_i \rangle$. The time evolution of $\langle n_i \rangle$ is governed by

$$\frac{d}{dt}\langle n_i \rangle = \sum_j \left\{ R_{ij}\langle n_j(1 + n_i) \rangle - R_{ji}\langle n_i(1 + n_j) \rangle \right\}.$$

This set of equations is not closed as the single-particle correlations depend on two-particle correlations, which in turn depend on three-particle correlations, and so on. To get a closed set of equations, we employ the mean-field approximation $\langle n_i n_j \rangle \simeq \langle n_i \rangle \langle n_j \rangle$, which then leads to

$$\frac{d}{dt}\langle n_i \rangle \approx \sum_j \left\{ R_{ij}\langle n_j \rangle[1 + \langle n_i \rangle] - R_{ji}\langle n_i \rangle[1 + \langle n_j \rangle] \right\}.$$

For the steady state, we have

$$\sum_j \left\{ R_{ij}\langle n_j \rangle[1 + \langle n_i \rangle] - R_{ji}\langle n_i \rangle[1 + \langle n_j \rangle] \right\} = 0. \tag{35}$$

In order to check the validity of the mean-field results, we perform a semi-classical Monte-Carlo simulation [5]. In this approach, the density matrix is approximated by a mixed superposition of Fock states $\rho = \sum_{\mathbf{n}} p_{\mathbf{n}}|\mathbf{n}\rangle\langle\mathbf{n}|$, with $\mathbf{n} = (n_1, n_2, \ldots, n_M)$, i.e., the off-diagonal elements which decouple with the diagonal elements and decay with time are neglected for weak system-bath coupling [75]. The equations of motion for the Fock-space occupation probabilities $p_{\mathbf{n}}$ is then mapped to a random walk in the classical space spanned by the Fock states $|\mathbf{n}\rangle$ (not their superposition) [5]. This method gives accurate results after sufficient statistical sampling.

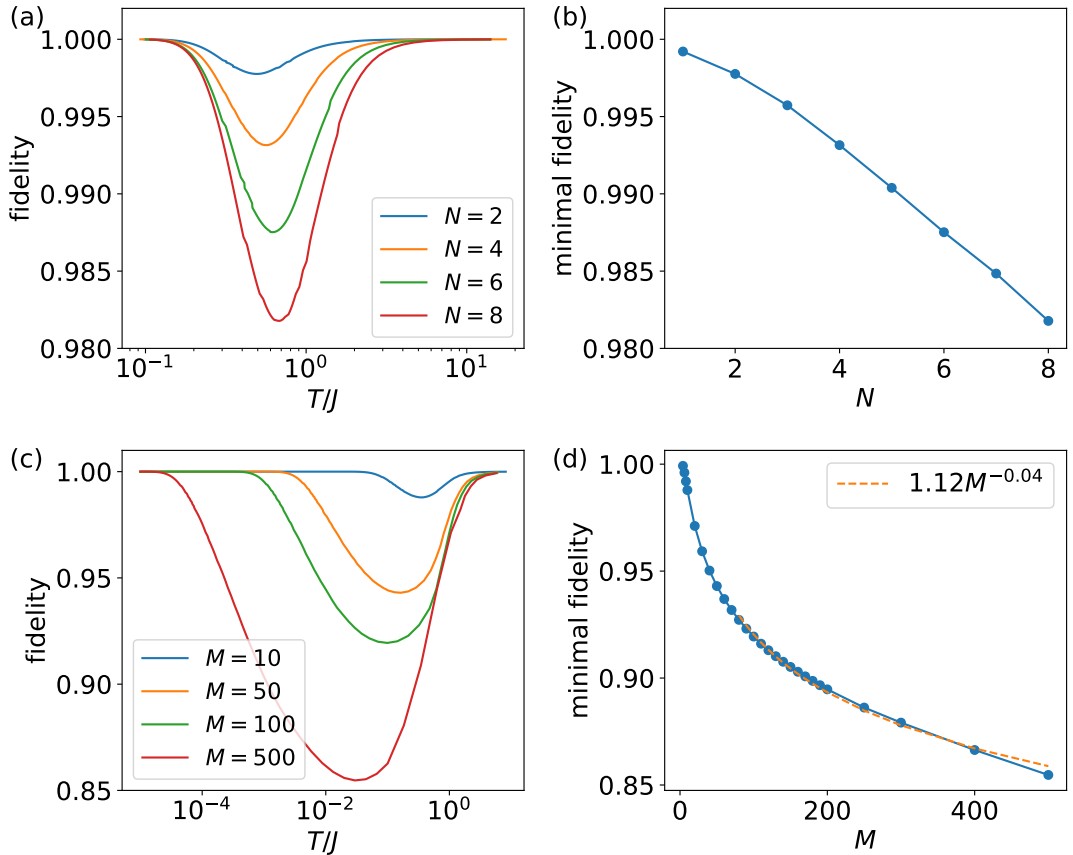

Figure 4: (a) The fidelity between the steady state and the corresponding thermal state as a function of $T$ for various $N$. (b) The minimal fidelity vs $N$. (c) The fidelity between the steady state and the corresponding thermal state as a function of $T$ for various $M$. (d) The minimal fidelity vs $M$. The parameters are $M = 4$ for (a) and (b), $N = 1$ for (c) and (d), $U = 0$, and $\gamma = 0.001J$.

In Fig. 5, we compare the mean-field results of Eq. (35) (dashed lines) with the exact results (solid lines) and the semi-classical Monte-Carlo results (circles). For a small system with $M = 4$ and $N = 8$ [see Fig. 5(a)], there is a tiny deviation between the mean-field results and the other two (which show excellent agreement with each other). For a larger system with $M = 10$ and $N = 50$ [see Fig. 5(b)], which is not accessible by exact diagonalization, we find a good agreement between the mean-field results and that from semi-classical Monte-Carlo simulations. These results give us the confidence to use Eq. (35) to study larger systems in the following.

If the system is coupled to a thermal bath of temperature $T$, $\langle n_i \rangle$ should satisfy the Bose distribution,

$$\langle n_i \rangle_T = \frac{1}{e^{(E_i - \mu)/T} - 1}, \tag{36}$$

where the chemical potential $\mu$ is fixed by $\sum_i \langle n_i \rangle_T = N$. In Fig. 6, we compare the results of Eqs. (35) [solid lines in Figs. 6(a), (d)] and (36) [dashed lines in Figs. 6(a), (d)] for $M = 6$ and $N = 100$. Here, the temperature $T$ [see Fig. 6(b)] is optimized to give the minimal error [see Fig. 6(c)]

$$\varepsilon = \frac{1}{N} \sqrt{\sum_i (\langle n_i \rangle - \langle n_i \rangle_T)^2}, \tag{37}$$

which is an intensive quantity. We can see that over the whole parameter regime, the two dis-

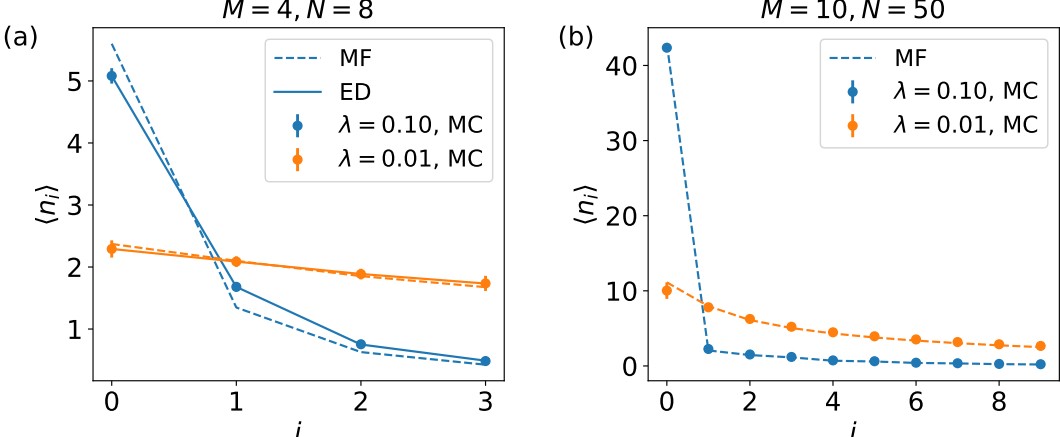

Figure 5: Comparison of the mean occupations in the single-particle eigenstate $\langle n_i \rangle$ between Monte-Carlo (MC), mean-field (MF) and exact diagonalization (ED) results. The MC results are obtained by averaging over 1000 trajectories. The error bars denote one standard deviation. Most of the time, they are smaller than the bullets.

tributions agree with each other very well. The maximal error is found at some intermediate temperature regime, marked by black dotted line in Fig. 6(c). Even in this case, the disagreement between the two distribution is very small, as shown by the black curves in Fig. 6(d).

Note that in thermodynamic limit, $M \rightarrow \infty$ at constant density $N/M$, thermal fluctuations prevent the formation of a Bose condensate in a one-dimensional system at finite temperature. However, for a finite size system, a crossover into a Bose condensed regime with a relative occupation of order 1 in the ground state occurs when the temperature $T$ reaches the condensation temperature $T_c \approx 8.3NJ/M^2$ [7], which is defined as the temperature where half of the particles occupy the single-particle ground state. By inspecting the mean occupations in the energy eigenbasis as shown in Figs. 6(a), (d), one can see that the error slightly increases in the crossover regime, where a condensate builds up. Note that in the limit of infinite temperature the feedback strength $\lambda$ approaches zero [see orange dotted line in Fig. 6(b)], so that the system is only subjected to continuous measurement, which is known to guide the system into an infinite temperature state.

Figure 7 shows the dependence of the error on the lattice size. In Fig. 7(a), we fix the filling factor at $N/M = 5$, while in Fig. 7(b) we fix the particle number $N$. In both cases, the error increases with the system size. Although the maximal error becomes significant for large systems, the error at low temperatures (for instance $T \simeq 0.1J$ as shown in dashed lines) is still small.

# 5 Interacting case

So far, we have discussed about the non-interacting cases. Now let us consider interacting particles. Figure 8 shows the results for $M = N = 4$ with interaction strength $U = 4J$. We can see that over a wide parameter regime, a high fidelity is still available. It is remarkable to observe that by solely employing measurement and feedback operators that are quadratic in the field operators (i.e. single-particle operators) it is sufficient to engineer thermal baths for an interacting system with high fidelity. Different from the non-interacting case, where the minimal fidelity is found at an intermediate temperature $T$, here the fidelity increases monotonously with $T$, as shown in Fig. 8(b). While even in the worst case, the distribution of

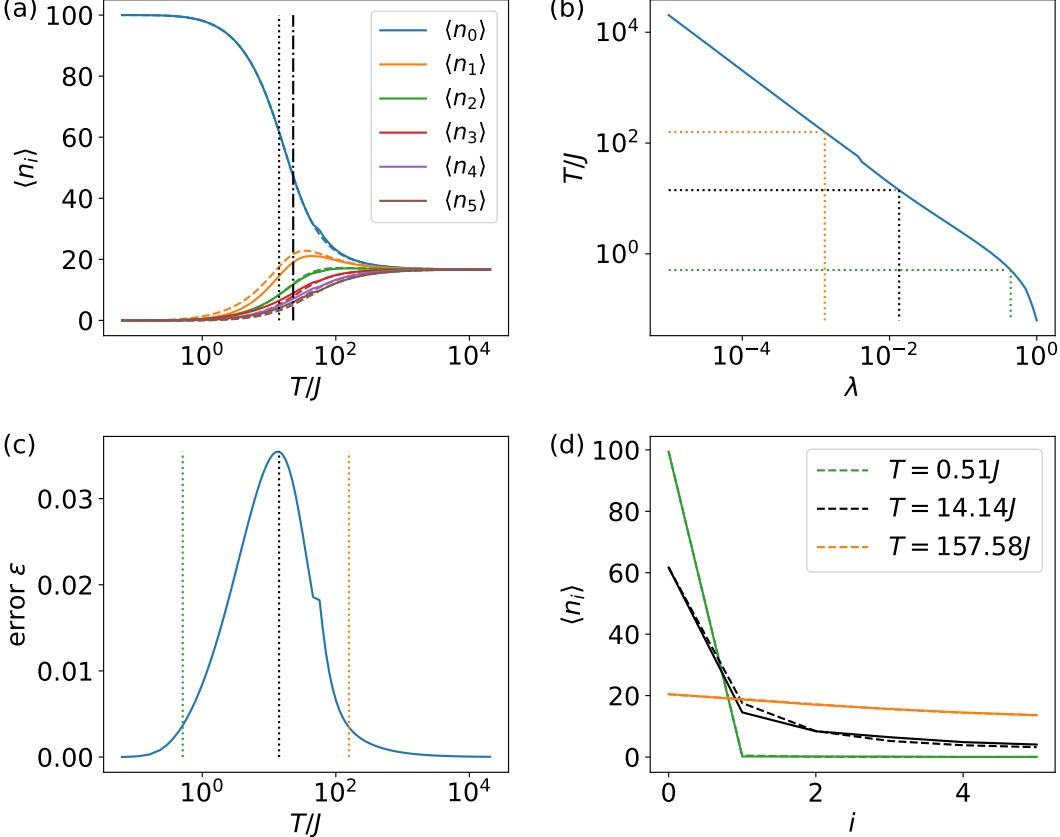

Figure 6: (a) Comparison of the results from Eqs. (35) (solid lines) to a thermal distribution (36) (dashed lines), whose corresponding temperature $T$ as a function of $\lambda$ is shown in (b). $T$ is fixed by minimizing the error (37), which is shown in (c) as a function of $T$. (d) Comparison of $\langle n_i \rangle$ (solid lines) and $\langle n_i \rangle_T$ (dashed lines) at three different $T$ marked by dotted lines in (b) and (c). In (a), the vertical dotted line marks the temperature where the maximal error occurs; the vertical dot-dashed line marks the condensation temperature $T_c \approx 8.3 NJ/M^2$ [7], which is defined as the temperature where half of the particles occupy the single-particle ground state. The parameters are $M = 6$, $N = 100$.

the steady state in the eigenbasis is found to be close to a thermal distribution, as shown by the blue curves in Fig. 8(d).

For the interacting case, the minimal temperature of the effective thermal bath that can be engineered is nonzero, and increases roughly linearly with the interaction strength $U$, as shown in Fig. 9(a). Given that a zero-temperature bath would lead to the ground state as steady state, the increase of the achievable bath temperature with interaction strength seems to imply worse performance of our scheme in preparing the ground state also for strong interactions. But this is not the case, as shown in Fig. 9(b). The fidelity between the steady state and the ground state (orange dot-dashed line) is found to first drop with increasing interaction strength, and then starts to gradually increase after reaching a dip. Over the whole parameter regime, the fidelity is considerably high, noting that the fidelity per particle is over $0.87^{1/N} \approx 0.97$ for $N = 4$ particles. This is attributed to the fact that for the strongly interacting case, the energy gap between the ground state and the first excited state is of the order of $U$, and thus also increases with the interaction strength. Hence, despite the increase of achievable bath temperature with increasing interaction strength, our scheme performs well

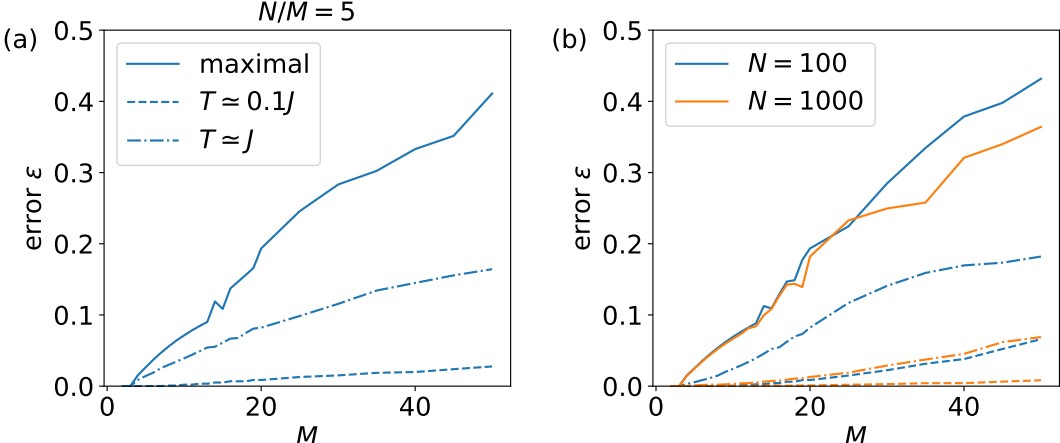

Figure 7: Error (37) as a function of lattice site $M$. In (a), the filling factor is fixed at $N/M = 5$. In (b), the particle number $N$ is fixed. The solid lines are the maximal errors, the dashed lines are the errors at $T \simeq 0.1J$, and the dot-dashed lines are the errors at $T \simeq J$.

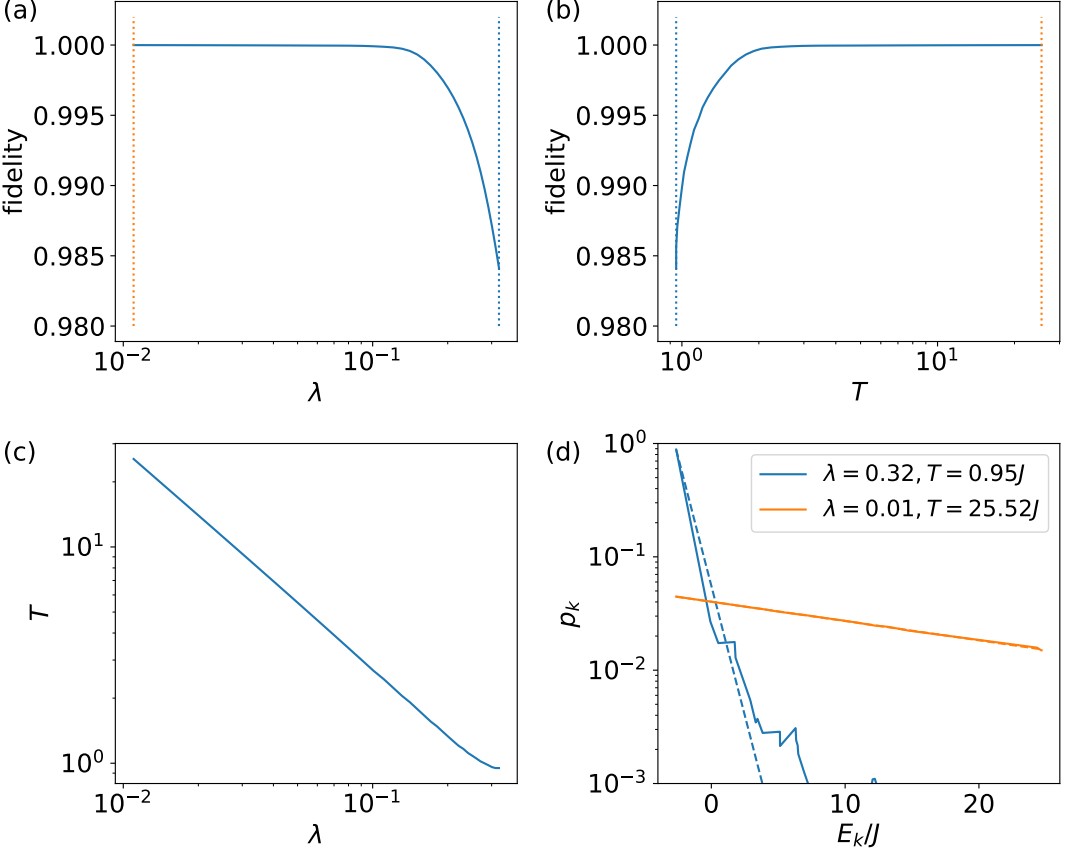

Figure 8: The fidelity between the steady state and the corresponding thermal state as a function of (a) the free parameter $\lambda$ and (b) the temperature $T$. The temperature of the thermal state is fixed by optimizing the fidelity and is shown in (c). (d) compares the distribution of the steady state (solid lines) and the corresponding thermal state (dashed lines) on the eigenbasis of the system at two different $\lambda$ [marked by dashed lines in (a) and (b)]. The parameters are $M = 4$, $N = 4$, $U = 4J$, $\gamma = 0.01J$.

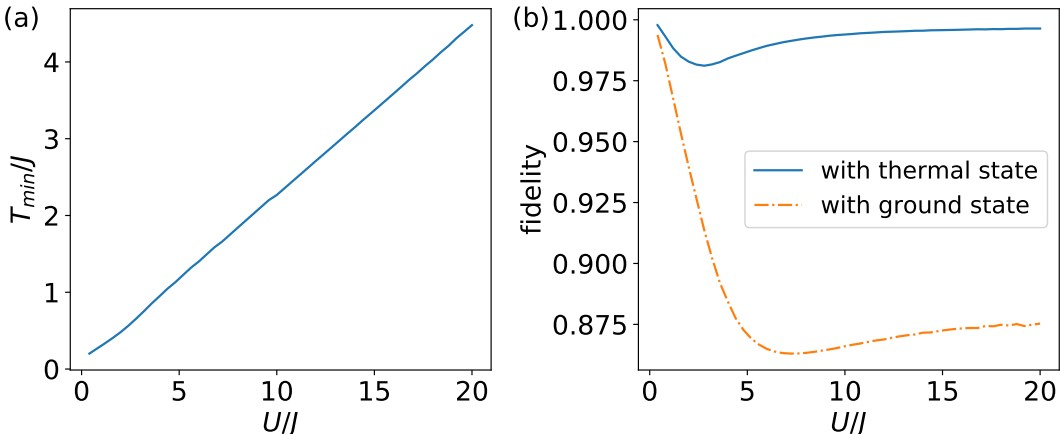

Figure 9: (a) The minimal temperature of the effective thermal bath that can be engineered as a function of the interaction strength. (b) The fidelity between the steady state and the corresponding thermal state (blue solid line) and the ground state (orange dot-dashed line). The parameters are $M = 4$, $N = 4$, $\gamma = 0.01J$.

both at finite temperatures and for the preparation of the ground state.

# 6 Conclusion

In conclusion, we have shown that Markovian feedback control can be used to engineer a synthetic thermal bath for a bosonic quantum gas in a one-dimensional optical lattice. For small systems, our discussions are focused on the steady state of the Markovian feedback ME (6), which is obtained from exact diagonalization of the Liouvillian superoperator. The performance of our scheme is quantified by the fidelity between the steady state and the effective thermal state. For double-well and triple-well systems with non-interacting particles, the transfer rates between all pairs of coupled eigenstates satisfy detailed balance condition, and thus the steady state is a thermal state. The scenario changes when there are more lattice sites, where the detailed balance condition does not hold any more, but also here deviations from detailed balance remain small. Remarkably, our scheme performs very well at low and high temperature regimes, with the fidelity close to one. The performance at the intermediate temperature regime is slightly worse, and the fidelity shows a gentle decrease with increasing system size. Due to the rapid growth of the dimension of the Hilbert space, it is time and memory consuming to treat large systems by using exact diagonalization. We then use kinetic theory to study the mean occupations in the single-particle eigenstate (35), and compare them with the Bose distribution (36). The results for large systems show similar behavior with those for small ones. Disagreement between the distributions appears at intermediate temperatures, where a crossover into a Bose condensed regime occurs. All the above discussions are for non-interacting systems. For the interacting cases, the scheme is found to perform better at a higher temperature, in contrast to the non-interacting cases. Moreover, the minimal temperature that can be engineered is nonzero and increases with the interaction strength. Nevertheless, the performance of our scheme is found to be good over the whole parameter regime, from weak interactions to strong interactions. As an outlook, it will be interesting to engineer driven dissipative systems via feedback control and study the nonequilibrium dynamics or steady states.

## Acknowledgements

**Funding information** This research was funded by the German Research Foundation (DFG) within the collaborative research center (SFB) 910 under project number 163436311.

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
