# Peer review of "Quantum engineering of a synthetic thermal bath for bosonic atoms in a one-dimensional optical lattice via Markovian feedback control"

_SciPost Physics, doi:SciPost Phys. 13, 059 (2022)_

## Round 1 · Referee Report · Anonymous (Referee 1) · 2022-4-28

Strengths

1 - The paper is well contextualized and acknowledges the work leading to it. 2- There is clarity in the derivation of the analytical results and good understanding of the numerical results. 3- There is a good progression on their analysis working from small towards large system sizes, helping to build intuition for the reader.

Weaknesses

1- The results are applicable in a restricted range of system sizes and no competing mechanisms working against the preparation scheme were considered. 2- The paper lacks a discussion of the required experimental parameters that would allow to observe this in a current platform.

Report

In this manuscript, the authors have discussed the use of a measurement and feedback scheme for dissipative state preparation of a thermal state with arbitrary temperature. They relay on the link between the state transfer rate, dependent solely on the form of the feedback, and the detailed balance condition to establish a correspondence between their complex tunneling phase lambda and the temperature. Although this relation is only exact for small systems, the authors discuss the validity of their scheme for less idealistic conditions and larger systems.

I consider that this manuscript includes significant contribution that would be of interest to the community. Thus, I recommend it for publication in the current form. Nonetheless, I would like that the authors address a few points indicated below, that would hopefully contribute to the clarity of the paper.

Requested changes

1- The authors discussed the form of the measurement in Eq. (8), (9) which is spatially modulated. It would be helpful to include a depiction of the z_l profile in this part for the visualization of the operator form. 2- In Sec. 2, the authors motivate the focus on Markovian, i.e. instantaneous, feedback and later on discuss the possibility of using an optical cavity to enhance the detection efficiency. Along these lines, it could be nice to see a small part in Sec. 2 or 3 devoted to discussing the realistic parameters required to meet on a current neutral atom experiment. 3- When using the semi-classical Montecarlo method, the authors include the results for 1000 trajectories. Are the error bars smaller than the lines? 4- Both in the discussion of the interactive case and in the conclusion the authors mentioned that the preparation scheme works well for the case of high T. However, as this is simply the case where the feedback is zero and thus we are only heating the system by continuous measurement, as mentioned by the authors in page 5, I would remind the reader of this fact when describing the good performance. 5- In a similar way in the regime of no feedback, it would be interesting to see the fidelity in the presence of a competing dissipative source that reduces the triviality of the regime. But I understand this can be the focus on a future publication.

  • validity: high
  • significance: high
  • originality: high
  • clarity: high
  • formatting: excellent
  • grammar: perfect

Author:  Lingna Wu  on 2022-05-12  [id 2461]

(in reply to Report 1 on 2022-04-28)

We are happy about the positive feedback from the referee and thank them for recommending the publication of our work. We thank the referee also for their useful comments, which we address in detail below.

Comment 1:
The authors discussed the form of the measurement in Eq. (8), (9) which is spatially modulated. It would be helpful to include a depiction of the z_l profile in this part for the visualization of the operator form.

Reply:
We thank the referee for this comment. A figure (Fig. 1 in the revised manuscript) has been added as suggested.

Comment 2:
In Sec. 2, the authors motivate the focus on Markovian, i.e. instantaneous, feedback and later on discuss the possibility of using an optical cavity to enhance the detection efficiency. Along these lines, it could be nice to see a small part in Sec. 2 or 3 devoted to discussing the realistic parameters required to meet on a current neutral atom experiment.

Reply:
We have added a sentence “For cold atoms in cavity (a potential experimental setup to implement our scheme), the typical time scales (such as tunneling time) are on the order of milliseconds [69]. Hence, a control on the higher kHz scale is sufficient, which can be achieved easily using digital signal processors.” after the sentence “Such a feedback is assumed to be instantaneous, namely, the delay time between the measurement and the application of the control field is small compared to the typical timescales of the system.” on page 3.

Comment 3:
When using the semi-classical Montecarlo method, the authors include the results for 1000 trajectories. Are the error bars smaller than the lines?

Reply:
The error bars were not shown in the plot. We have now included error bars in the new plot.

Comment 4:
Both in the discussion of the interactive case and in the conclusion the authors mentioned that the preparation scheme works well for the case of high T. However, as this is simply the case where the feedback is zero and thus we are only heating the system by continuous measurement, as mentioned by the authors in page 5, I would remind the reader of this fact when describing the good performance.

Reply:
We thank the referee for this comment. As suggested, we have added a sentence “Note that in the limit of infinite temperature the feedback strength $\lambda$ approaches zero [see orange dotted line in Fig. 6(b)], so that the system is only subjected to continuous measurement, which is known to guide the system into an infinite temperature state.” in the discussion of Fig. 6.

Comment 5:
In a similar way in the regime of no feedback, it would be interesting to see the fidelity in the presence of a competing dissipative source that reduces the triviality of the regime. But I understand this can be the focus on a future publication.

Reply:
We totally agree with the referee that it would be interesting to introduce other dissipative sources and investigate the nonequilibrium dynamics or steady state of the system. While the manuscript is already quite long, and we want to focus on the quantum engineering of thermal bath based on feedback control. Hence, we decide not to include detailed discussions on this point, but add an outlook in the conclusion paragraph.

---

## Round 1 · Referee Report · Santiago Francisco Caballero Benitez (Referee 2) · 2022-4-29

Strengths

1)The article is clear and well written.
2)Several different techniques and approximation schemes are explored.
3)Comparison of different approaches used in the paper is given.
4)The manuscript innovates and gives a fresh take on some explored ideas.
5)It uses tools of different areas to give a new path of research.

Weaknesses

1)Title is a bit misleading.
2)The scope of the article should be clearly communicated.
3)Relation with physical observables is lacking.
4)Relevant physical discussion of some phenomenology expected is lacking.

Report

The manuscript "Quantum engineering of a synthetic thermal bath via Markovian feedback control" by wu and Eckardt, shows how to engineer thermal states for non-interacting bosons in a 1D lattice mainly. The manuscript is well written and an extensive comparison of the different results for small systems is given while a mean-field approach is employed for longer chains.The computations are sound. The title of the article is a bit misleading as the authors have a very specific case and moreover they concentrate on the non-interacting cases. The results in general are interesting and it should be useful a basis to continue further research in the area. The results are not particularly surprising as the there is a body of work regarding engineering of thermal states. However the novelty of using the Markovian feedback is interesting and refreshing. I think the article could be considered further once some concerns and clarifications are made.

Requested changes

My concerns are:

1)The title of the article is a bit misleading, I suggest to change it and include the fact that the authors are working with bosons in 1D and non-interacting for the most part.

2)I suggest to change equation (1) to the non interacting case and later introduce the interaction in the section devoted to it, while modifying the captions adequately.

3)The discussion regarding the interacting case needs to be extended. The regime where the authors are considering the system is still weakly interacting with respect to insulating transitions, a discussion of this should be made.

4)In figure 6, the maximum error reported is the absolute error or the relative error.

5)As the temperature increases there should be evidence of the transition between the normal state and the BEC for the non-interacting case while between SF-Normal with interactions. Do the authors recover the expected behaviour of this in some observable?

6)It would be desirable to have the information of some observable like number fluctuations in addition to information of the distribution, to have a clearer physical picture and relate it with accesible experimental quantities.

  • validity: good
  • significance: good
  • originality: high
  • clarity: top
  • formatting: perfect
  • grammar: perfect

Author:  Lingna Wu  on 2022-05-12  [id 2462]

(in reply to Report 2 by Santiago Francisco Caballero Benitez on 2022-04-29)

We are happy that the referee rates our paper well written and finds our results convincing. We thank the referee also for his useful comments, which we address in detail below.

Comment 1:
The title of the article is a bit misleading, I suggest to change it and include the fact that the authors are working with bosons in 1D and non-interacting for the most part.

Reply:
We thank the referee for the suggestion and have changed the title to “Quantum engineering of a synthetic thermal bath for bosonic atoms in a one-dimensional optical lattice via Markovian feedback control”.

Comment 2:
I suggest to change equation (1) to the non interacting case and later introduce the interaction in the section devoted to it, while modifying the captions adequately.

Reply:
Given that both non-interacting and interacting cases are discussed (although the focus is on the non-interacting case), we would prefer to keep the structure of the manuscript. Namely, first introduce the general model, and then discuss the non-interacting and interacting cases, respectively, in the following sections.

Comment 3:
The discussion regarding the interacting case needs to be extended. The regime where the authors are considering the system is still weakly interacting with respect to insulating transitions, a discussion of this should be made.

Reply:
We thank the referee for raising this point and have now extended the discussion on the interacting case.
Regarding the interaction strength, actually we did consider the case of strong interactions (see Fig. 8 of the previous manuscript, or Fig. 9 of the revised manuscript). However, we did not discuss it in detail in the previous manuscript. To further stress the impact of interactions, we have raised the interaction strength of Fig. 7 of the previous manuscript (or Fig. 8 of the revised manuscript) from U=2J to U=4J. Moreover, we have added a new paragraph discussing the role of interactions.

Comment 4:
In figure 6, the maximum error reported is the absolute error or the relative error.

Reply:
It is the maximum value of the error defined in Eq. (37), which is an intensive quantity taking values between 0 and 1. To avoid potential confusion, we have added a sentence below Eq. (37) to stress it.

Comment 5:
As the temperature increases there should be evidence of the transition between the normal state and the BEC for the non-interacting case while between SF-Normal with interactions. Do the authors recover the expected behaviour of this in some observable?

Reply:
The referee raises an important point, which we did not discuss sufficiently in the previous manuscript. For the non-interacting case, the mean-field results show indeed the expected behavior (see old Fig. 5 or new Fig. 6). We have added a new paragraph discussing the crossover to a finite-size Bose condensate on page 13, as copied here:
“Note that in thermodynamic limit, $M \rightarrow \infty$ at constant density $N/M$, thermal fluctuations prevent the formation of a Bose condensate in a one-dimensional system at finite temperature. However, for a finite size system, a crossover into a Bose condensed regime with a relative occupation of order 1 in the ground state occurs when the temperature $T$ reaches the condensation temperature $T_{c} \approx 8.3NJ/M^2$ [7], which is defined as the temperature where half of the particles occupy the single-particle ground state. By inspecting the mean occupations in the energy eigenbasis as shown in Figs. 6(a), (d), one can see that the error slightly increases in the crossover regime, where a condensate builds up. Note that in the limit of infinite temperature the feedback strength $\lambda$ approaches zero [see orange dotted line in Fig. 6(b)], so that the system is only subjected to continuous measurement, which is known to guide the system into an infinite temperature state.”
For interacting cases, due to the computational limitations, we are not able to investigate sufficiently large systems.

Comment 6:
It would be desirable to have the information of some observable like number fluctuations in addition to information of the distribution, to have a clearer physical picture and relate it with accesible experimental quantities.

Reply:
We agree with the referee that it would be helpful to investigate some other observables. However, our study of large systems is based on the mean-field theory, which well describes the grand-canonical ideal gas. Since the condensate and total particle number fluctuations are known to deviate strongly between the grand canonical and the canonical regime (“grand canonical fluctuation catastrophe”), this theory is not able to accurately predict number fluctuations for the system of fixed total particle number. (Nevertheless, it provides accurate results for the mean occupations.) Therefore, accurately computing number fluctuations would require different methods, which is beyond the scope of the present work.

---

## Round 2 · Referee Report · Santiago Francisco Caballero Benitez (Referee 2) · 2022-5-18

Report

The authors have answered all my concerns and have improved the manuscript. The new additions to the interacting part improved the discussion. Therfore, I recommend the article to be published as it is in Scipost Physics.

---

## Round 2 · Referee Report · Anonymous (Referee 1) · 2022-5-26

Strengths

1 - The paper is well contextualized and acknowledges the work leading to it.
2- There is clarity in the derivation of the analytical results and good understanding of the numerical results.

3 - There is a good progression on their analysis working from small towards large system sizes, helping to build intuition for the reader.
4 - Experimental feasibility, even if briefly, is discussed.

Weaknesses

1- The results are applicable in a restricted range of system sizes and no competing mechanisms working against the preparation scheme were considered.

Report

I believe that the authors - by providing additional figures, explanations and outlook - have addressed all the proposed changes. As a result, I believe that the article deserves publication in its current form.

Requested changes

No further changes required.

---

## Round 2 · Author Response

Dear Editor,

With this letter we resubmit our manuscript “Quantum engineering of a synthetic thermal bath for bosonic atoms in a one-dimensional optical lattice via Markovian feedback control” for publication in SciPost Physics.

We thank both referees for their professional criticism and useful comments, which allowed us to improve our manuscript considerably.

We hope the revised manuscript will live up to the standards of SciPost Physics and can be accepted for publication.

Thank you!

Sincerely yours,
Ling-Na Wu and André Eckardt

---

## Round 2 · List of Changes

1. The title has been changed from “Quantum engineering of a synthetic thermal bath via Markovian feedback control” to “Quantum engineering of a synthetic thermal bath for bosonic atoms in a one-dimensional optical lattice via Markovian feedback control”, as a response to Referee #2 (comment #1).

  2. In the abstract, a sentence has been added: “(where a crossover into a Bose condensed regime occurs)”.

  3. On page 3, a sentence “For cold atoms in cavity (which is a potential experimental setup to implement our scheme), the typical time scales (such as tunneling time) are on the order of milliseconds [69]. Hence, a control on the higher kHz scale is sufficient, which can be achieved easily using digital signal processors.” has been added to address comment #2 from Referee #1.

  4. A figure (Fig. 1) has been added, as suggested by Referee #1 (see comment #1).

  5. In Fig. 5, errorbars are added, as a response to Referee #1 (see comment #3).

  6. On page 13, under Eq. (37), a sentence “which is an intensive quantity taking values between 0 and 1.” has been added to address comment # 4 from Referee #2.

  7. On page 13, a new paragraph discussing the crossover to a finite-size Bose condensate has been added to address comment #5 from Referee #2. The last sentence is added as suggested by Referee #1 (comment #4).

  8. The discussion of the interacting case (Section 5) has been extended, as suggested by Referee #2 (comment #3).

  9. The interaction strength for Fig. 8 has been raised from 2J to 4J, to further stress the impact of interactions.

  10. In Fig. 9 (b), a curve denoting the fidelity between the steady state and the ground state has been added.

  11. The conclusion paragraph has been extended, including an outlook (as a response to comment #5 from Referee #1).

---

## Editorial Decision

published